# Strong Sex-Dependent Effects of Malnutrition on Life- and Healthspan in *Drosophila melanogaster*

**DOI:** 10.3390/insects15010009

**Published:** 2023-12-26

**Authors:** Nikolaj Klausholt Bak, Palle Duun Rohde, Torsten Nygaard Kristensen

**Affiliations:** 1Department of Chemistry and Bioscience, Aalborg University, Frederik Bajers Vej 7H, DK 9220 Aalborg, Denmark; ndb@bio.aau.dk; 2Department of Health Science and Technology, Aalborg University, Selma Lagerløfs Vej 249, DK 9260 Gistrup, Denmark

**Keywords:** *Drosophila melanogaster*, malnutrition, ageing, sex-dependent responses

## Abstract

**Simple Summary:**

Millions of people suffer from malnutrition worldwide, and not enough or suboptimal nutrients also constitute a major challenge for many other species. Here, we investigated the effects of undernutrition on phenotypes associated with the aging processes utilizing *Drosophila melanogaster* as a model species. We fed flies with either a balanced and nutritious diet or restricted diets with nutritional deficiencies. We found that flies fed on poor diets weighed less, had shortened lives, and were less active and more sensitive to environmental stress compared to flies fed on more nutritious diets. Interestingly, these negative impacts of undernutrition were much more severe in males than in females. We conclude that the period of life spent in good health conditions, i.e., healthspan, can be markedly reduced by nutritional stress and that female flies are better able to cope with an insufficient diet, possibly because of a larger innate fat deposit compared to males.

**Abstract:**

Insufficient intake of essential nutrients, malnutrition is a major issue for millions of people and has a strong impact on the distribution and abundance of species in nature. In this study, we investigated the effect of malnutrition on several fitness components in the vinegar fly *Drosophila melanogaster*. Four diets with different nutritional values, including three diluted diets of an optimal nutritional balanced diet, were used as feed sources. The effect of malnutrition on fitness components linked to healthspan, the period of life spent in good health conditions, was evaluated by quantifying the flies’ lifespan, locomotor activity, heat stress tolerance, lipid content, and dry weight. The results showed that malnutrition had severe negative impact, such as reduced lifespan, locomotor activity, heat stress tolerance, fat content, and dry weight. The negative phenotypic effects were highly sex-dependent, with males being more negatively impacted by malnutrition compared to females. These findings highlight important detrimental and sex-specific effects of malnutrition not only on lifespan but also on traits related to healthspan.

## 1. Introduction

Aging is a complex process that is important for all living organisms. It is characterized by a progressive decline in physiological function, which leads to increased susceptibility to diseases and, eventually, death. There is a growing interest in understanding the mechanisms underlying aging and developing interventions to delay or prevent age-related diseases. One of the promising interventions is dietary restriction (DR), which involves reducing caloric intake without causing malnutrition. DR has been shown to improve lifespan in a wide range of species, including yeast, nematodes, mice, and lemurs [1,2,3,4]. While the beneficial effects of DR on lifespan and physiological functioning at old age have been extensively studied, the potential negative effects of malnutrition have received less attention. Malnutrition, defined as a deficiency or excess of one or more essential nutrients, can have serious consequences for human health and well-being and be an important seasonal or permanent stress factor for natural populations [5,6]. One type of malnutrition is undernutrition, which is a major public health issue affecting over 900 million people worldwide and is responsible for 3.5 million deaths of children under the age of five [7]. Malnutrition may accelerate the decline in physiological processes with age, increase the risk of age-related diseases, cause premature death, and reduce physical performance and well-being at old age [5,8,9]. Additionally, cachexia, anorexia, malabsorption, and hypermetabolism are syndromes that can contribute to undernutrition in elderly persons, with cachexia being a severe disorder characterized by loss of muscle mass, and anorexia being a decrease in taste and olfaction, gastric emptying, and early satiation signals [10,11,12]. Nutritional stress, often referred to as starvation in natural populations, is important for the distribution and abundance of species [13], and starvation is a significant predictor of, e.g., historical extinctions of large mammals [14]. Recent examples show that nutritional low-quality leaves induce strong physiological stress responses in, e.g., red colobus monkeys (*Procolobus rufomitratus*), and insufficient nutrients during reproduction in long-lived seabirds have marked negative consequences for both survival and reproduction [15,16]. Thus, studying the effects of nutritional stress is of general relevance across species and research areas.

Lifespan is typically used as a measure of aging. However, it only measures the length of an organism’s life and does not consider the quality of life, i.e., the period when an organism is healthy and free from chronic disease or disability. Healthspan, on the other hand, encompasses the quality of life during the period when an individual is healthy and free from chronic disease or disability [17,18,19]. Therefore, it is important to investigate the effects of malnutrition on both lifespan and healthspan to obtain a more comprehensive understanding of the effects of malnutrition on the aging process. Predictive factors of healthspan in humans and many animals include physical functions: gait speed; cardiorespiratory fitness; and grip strength, as a reduction in these will increase disability of an individual [17,19]. In humans, a decline in locomotor activity has been seen with advancing age. Similar effects have been seen across many different species, such as fruit flies, nematodes, mice, rats, rhesus monkeys, and beagle dogs [20,21,22,23,24]. Furthermore, physiological functioning also includes robustness towards stressors such as heat or cold shock and oxidative stress, of which a higher tolerance to these stresses corresponds to improved healthspan [22,25].

In many species, including humans and *D. melanogaster*, females typically live longer than males [26,27]. Despite females living, on average, longer, they often live with health challenges for a longer part of their life than males [26,28]. These sex-specific disparities extend to various health conditions, including cancer and strokes, highlighting the need for more research on the topic [26,28,29]. Given these complexities, it is crucial to investigate how sex differences extend to various aspects of health, including the impacts of nutrition and malnutrition.

It can be challenging to study healthspan in humans and in species in nature due to the complexity and multifaceted character of traits associated with healthspan as healthspan is affected by a wide range of factors, including genetic variation, lifestyle, and environmental exposures [26,30,31,32]. One way to study healthspan under controlled environmental conditions is by using model organisms, such as mice or fruit flies. In this study, *D. melanogaster* was used as a model organism to study the effect of undernutrition on health- and lifespan. Using *D. melanogaster* in aging studies offers advantages linked to its relatively short lifespan, ease of handling in the laboratory, short generation intervals, limited ethical restrictions, and a well-characterized genome [33,34,35]. *D. melanogaster* also shares many conserved aging-related mechanisms and molecular pathways with humans, i.e., the *D. melanogaster* genome shares 65% of human disease genes, such as insulin/IGF-1 signaling, mTOR signaling, and oxidative stress responses [35,36].

*D. melanogaster* studies have previously shown that moderate reductions in dietary nutrients can lead to an increase in lifespan, while further reductions in dietary nutrients lead to a decrease in lifespan [37,38,39,40,41,42,43]. Furthermore, with a further reduction in nutrient intake, males have been shown to have a more pronounced decrease in lifespan compared to females [44]. However, the effect of undernutrition on aging might not only impact lifespan but also healthspan, which has been less investigated in *D. melanogaster*. In this study, the effects of undernutrition during the adult life stage were investigated on both longevity and multiple traits related to healthspan by measuring the effects on locomotor activity, heat stress tolerance, fat content, and dry weight of female and male *D. melanogaster*. Flies were exposed to four different diets, including a balanced control diet (100% diet) and three diets reduced in nutrients (with 50%, 25%, or 10% of the nutritional content in the control diet). These diets with reduced nutrients were achieved by adding α-cellulose, which is an indigestible compound, to the control diet.

We hypothesize a small increase in lifespan and traits related to healthspan (locomotor activity, heat stress tolerance, fat content, and dry weight) with a moderate reduction in nutrients followed by a decrease in lifespan and healthspan with a further reduction in nutrients in the diets [5,37,38,39,40,41,42,43,45]. Furthermore, we expect that the two sexes will respond differently to malnutrition, with males having a more pronounced decrease in lifespan and healthspan compared to females, when reared on diets with reduced nutritional value [44].

## 2. Materials and Methods

### 2.1. Fly Population and Nutritional Environments

The mass-bred population of *D. melanogaster* used in this study was founded by approximately 600 inseminated females caught in Odder (55°56′42.46″ N, 10°12′45.31″ E), Denmark, in October 2010. Flies were maintained at 20 °C and 50% RH at a 12:12 h light/dark cycle. Prior to the experiments, flies were reared on standard Leeds medium (also entitled control diet in the current study) composed of dry yeast (60 g L^−1^), sucrose (40 g L^−1^), oatmeal (30 g L^−1^), agar (16 g L^−1^), Nipagin (12 mL L^−1^) (Nipagin, Sigma-Aldrich, St. Louis, MO, USA), and acetic acid (1.2 mL L^−1^).

Four different nutritional diets were constructed. Apart from the control diet, we made diets with 50%, 25%, and 10% of the nutritional content of the control diet using the indigestible compound α-cellulose (Product no. 102550125, Sigma-Aldrich, Buchs, SG, Switzerland) (Table 1). The concentrations of agar, nipagin, and acetic acid were the same in all four diet types.

### 2.2. Experimental Design

To produce the flies used in the experiments, 1000 flies (mixed sex) were distributed with 200 individuals in each of five bottles from the mass-bred population and kept at 23 °C for 24 h. Leeds medium sprinkled with dry yeast was provided to increase female egg production. Eggs were collected and distributed with 40 eggs in each of 100 vials with standard Leeds medium. Thus, during development, all flies were exposed to a similar nutritious diet (the Leeds control diet). At emergence, females and males were lightly anesthetized with CO_2_ and transferred to new vials containing ten same-sex virgin individuals per vial. Thus, ten individuals per sex were distributed into ten vials with 3 mL medium per diet type (Table 1). Flies were kept at 23 °C and 50% RH at a 12:12 h light/dark cycle for all the experiments unless otherwise stated.

### 2.3. Lifespan Assay

For each sex and diet type (Table 1), we had a total of ten biological replicates, each with ten flies, making a total of 400 male and 400 female flies for lifespan assessment. Flies were transferred to new vials with 3 mL of medium every second day. At each transfer, dead flies were registered and discarded. This process was continued until all flies on all four diets were dead.

### 2.4. Locomotor Activity Assay

Flies from two biological replicates, each with 10 flies, from each nutritional type and sex, were transferred to 5 mm polycarbonate tubes (TriKinetics Inc., Waltham, MA, USA) containing Leeds medium in one end and a pipe cleaner moistened in water in the other end. We aimed to test 20 flies per sex and diet type from the age classes at 4 days, 12 days, 27 days, and 45 days. However, flies reared on the 25% and 10% diets died young, and, therefore, individuals from the 25% diet were only included at age 4 days, 12 days, and 27 days, and individuals from the 10% diet only at age 4 days and 12 days (Table 2). Flies in the polycarbonate tubes were added to *Drosophila* Activity Monitors (DAM2 Activity Monitor, Trikinetics Inc., Waltham, MA, USA) by randomly assigning flies from each replicate to columns of the monitors. The monitors were placed in a climate chamber (Binder KB 400, Binder, Tuttlingen, Germany) at 23 °C, and the locomotor activity was monitored by recording the number of times individual flies passed a laser in the middle of the tube every ten seconds for six hours (3:00 to 9:00 p.m.). Temperature and humidity conditions were recorded every ten minutes (Appendix A).

### 2.5. Heat Stress Tolerance

Following the investigation of locomotor activity at 23 °C, flies were kept in the activity monitors and exposed to 39 °C in a climate chamber (Binder KB 400, Binder, Tutt-lingen, Germany), while their locomotor activity was measured every 10 s for 2 h (9:30 to 11:30 p.m.), at which time all had died. Time to death due to heat was assessed as the last time a fly was registered to have crossed the infrared beam in the middle of the tube. Heat knockdown time (HKDT) was calculated as the number of minutes between when the flies were initially exposed to 39 °C, until they died. Temperature and humidity conditions were monitored every 10 min (Appendix A). Individual flies used for locomotor activity and heat tolerance assessment were stored in Eppendorf tubes at −80 °C for later dry weight and lipid content assessments.

### 2.6. Dry Weight and Lipid Content

Following the examination of heat stress tolerance in the activity monitors, flies were dried for 48 h at 60 °C and weighed to the nearest 10 μg (Sartorius Quintix35-1S, Satorius, Göttingen, Germany) [46,47,48,49]. After dry weight measurements, lipids were removed by adding 1 mL of chloroform to each individual Eppendorf tube with one fly in each for 24 h, after which the supernatant was discarded; this step was repeated one time [46,47,48]. Flies were then dried for 24 h at 60 °C and weighed to the nearest 10 μg. Lipid content was calculated by subtracting the dry weight after lipid extraction from the dry weight obtained before lipid extraction. Long-term exposure (several days) to different temperatures can impact energy stores in *D. melanogaster* [50]. To verify that this did not impact weight assessments in our setup with a short exposure (2 h) to high temperatures, we performed a pilot study testing the dry weight of flies that have, or have not, been exposed to high-temperature stress for 2 h. The dry weight of the flies from the two groups was similar, and we, therefore, argue that using flies that have been exposed to heat stress for dry weight and lipid content measurements is justified (Appendix A).

### 2.7. Statistical Analysis

All figures and statistical analyses were performed in R (version 4.3.1) [51]. Multivariate Cox regression was used to determine the difference in lifespan with sex and diet as independent variables using the R package ‘Survival’ [52].

A rank-based inverse normal transformation (INT) [53,54] was used on the locomotor activity, HKDT, body weight, and lipid content data to approximate a Gaussian distribution.
(1)$$INT\left(u_{i}\right)=\Phi^{-1}\left(\frac{rank\left(u_{i}\right)-0.5}{n}\right)$$
where $\Phi^{-1}$ is the probit function, $rank\left(u_{i}\right)$ denotes the sample rank of $u_{i}$, and n is the number of samples.

Analysis of variance (ANOVA) was used to determine differences in body composition with dry weight and lipid content of the flies as the dependent variable (Y). The ANOVA included the independent variables sex ($X_{1}$), diet ($X_{2}$), age group ($X_{3}$), and all possible interaction effects.
(2)$$Y=b_{0}+b_{1}X_{1}+b_{2}X_{2}+b_{3}X_{3}+b_{4}X_{1}X_{2}+b_{5}X_{1}X_{3}+b_{6}X_{2}X_{3}+b_{7}X_{1}X_{2}X_{3}$$
which not only accounts for the additive effect of the variables (e.g., $b_{1}X_{1}+b_{2}X_{2}$), but also the interaction effect among them (e.g., $b_{3}X_{1}X_{2})$, where the X-matrices represents design matrices linking the estimated effects to observations and b denotes regression coefficients.

A Tukey HSD was used to investigate at which age the diet affected flies’ body composition with dry weight and lipid content as dependent variables and sex, diet, age group, and all possible interaction effects as independent variables.

For healthspan, ANOVA was used with locomotor activity and HKDT as dependent variables and sex, diet, and age group as independent variables. Additionally, a Tukey HSD was used to investigate at which age the diet affected flies’ healthspan with locomotor activity and HKDT as dependent variables and sex, diet, age group, and all possible interaction effects as independent variables.

Dry weight, lipid content, locomotor activity, and HKDT have been measured on the same individual flies, which means that the data are linked. Because the data have been transformed using the rank-based inverse normal transformation, this made it possible to sum the values for dry weight, lipid content, locomotor activity, and HKDT for every individual fly and then scale the data from 0 to 1.
(3)$$z_{i}=\frac{x_{i}-min\left(x\right)}{max\left(x\right)-min\left(x\right)}$$

For the fitness score, it is assumed for all traits that higher values constituted a fitness benefit. The fitness score ($\sum_{i=1}^{n}z_{i}$, with n being the number of traits) was subjected to ANOVA with sex, diet, and age group as independent variables. Furthermore, a Tukey HSD was used to investigate at which age the diet affected flies’ fitness score with sex, diet, age group, and all possible interaction effects as independent variables.

## 3. Results

### 3.1. Lifespan Assay

Across the four different diets, female flies survived, on average, 4% (5 days) longer than male flies (*p* = 0.016) (Figure 1 and Appendix A). Diet impacted significantly on lifespan in both females (*p* < 0.001) and males (*p* < 0.001). Female lifespans were markedly decreased on the 10 and 25% diets (Figure 1a), while male lifespan already decreased in flies exposed to the 50% diet (Figure 1b).

### 3.2. Dry Weight and Lipid Content

To investigate whether diet and sex affected body composition, an ANOVA test was performed with dry weight and lipid content of the flies as the dependent variables. Sex, diet, age group, and all possible interaction effects were included in the model, and results showed that there was a significant difference between sexes in dry weight (*p* < 0.001; Appendix A), with female flies having an average dry weight 25% (0.10 mg) higher than males. There was a significant interaction between sex and diet (*p* = 0.006; Appendix A) and among sex, diet, and age group (*p* = 0.027; Appendix A). The analysis further showed a significant interaction between diet and age group (*p* < 0.001; Appendix A), which was also seen when performing the analysis for each sex separately (Appendix A). In addition, diet and age also had a significant effect on dry weight when the sexes were separated (Appendix A). Female flies on the control diet at the age of 12 days had a 7% (0.03 mg) and 30% (0.14 mg) decrease in dry weight compared to the 25% diet and 10% diet, respectively (Figure 2a and Appendix A). Additionally, female flies on the 50% diet and 25% diet at the age of 12 days had a 28% (0.12 mg) and 25% (0.11 mg) decrease in dry weight, respectively, compared to the 10% diet (Figure 2a and Appendix A). Male flies on the control diet, 50% diet, and 25% diet at the age of 12 days had a 12% (0.04 mg), 14% (0.04 mg), and 11% (0.03 mg) decrease in dry weight, respectively, compared to the 10% diet (Figure 2b and Appendix A).

An ANOVA, which included sex, diet, age group, and all interaction effects among these, was performed to investigate whether lipid content could be explained by sex and diet. The results showed a significant difference in lipid content between female and male flies (*p* < 0.001, Appendix A), with females having 43% (0.04 mg) more lipid content than males (Figure 2c,d). There was also a significant interaction effect between sex and diet (*p* < 0.001, Appendix A) and among sex, diet, and age group (*p* < 0.001, Appendix A). Additionally, there was a significant effect of age for both sexes on lipid content (*p* < 0.001, Appendix A). Diet showed a significant effect on lipid content for females (*p* < 0.001, Appendix A) but no effect on males (*p* = 0.204, Appendix A). However, a significant interaction effect between diet and age was observed for males (*p* = 0.017, Appendix A). Female flies on the control diet, 25% diet, and 50% diet at the age of 12 days had an 83% (0.11 mg), 83% (0.10 mg), and 77% (0.07 mg) decrease in lipid content, respectively, compared to the 10% diet (Figure 2c and Appendix A).

### 3.3. Locomotor Activity Assay

We tested the influence of diets on a healthspan correlate, namely, locomotor activity. This test included sex, diet, age group, and all possible interaction effects. A significant difference between sexes in activity was observed (*p* < 0.001, Appendix A), with female flies having an average activity 15% (0.13 counts per minute) higher than males (Figure 3a,b). Significant interaction effects between sex and age group (*p* = 0.033, Appendix A) and among sex, age group, and diet were also found (*p* = 0.007, Appendix A). The analysis further revealed a significant effect of diet (*p* < 0.001, Appendix A), which was also detected when testing the sexes separately (Appendix A). Male flies showed a significant effect of age group (*p* = 0.019, Appendix A) and a significant interaction effect between diet and age group (*p* = 0.001, Appendix A).

Male flies on the control diet at the age of 12 days had a 63% (0.58 counts per minute) increase in activity compared to those on the 10% diet (*p* < 0.001, Figure 3b and Appendix A). Additionally, male flies on the 50% diet at the age of 12 days had a 67% (0.70 counts per minute) increase in activity compared to the 10% diet (*p* < 0.001, Figure 3b and Appendix A).

### 3.4. Heat Stress Tolerance

An ANOVA test was used to analyze the heat knockdown time (HKDT) of the flies. The model included sex, diet, age group, and all the possible interaction effects. Female and male flies differed significantly in HKDT, with male flies surviving 8.7% (3.8 min) longer than female flies (*p* < 0.001, Appendix A). The different age groups had different heat tolerance (*p* < 0.001, Appendix A), and there was also a significant interaction effect between age group and sex (*p* < 0.001, Appendix A). The analysis further showed a significant difference in heat tolerance among diets (*p* < 0.001, Appendix A), and the interaction effect including all variables—diet, age group, and sex—was significant (*p* = 0.017, Appendix A). There was a significant difference in heat tolerance between diet and age group of female flies (Appendix A). All the tested variables for male flies—diet, age group, and the interaction effect between diet and age group—were significant (Appendix A).

Female flies on the control diet, 50% diet, and 25% diet at the age of 12 days had a 34% (15.4 min), 28% (12.1 min), and 36% (16.9 min) decrease in HKDT, respectively, compared to the 10% diet (Figure 4a and Appendix A). Male flies on the control diet, 50% diet, and 25% diet at the age of 12 days had a 26% (13.3 min), 19% (8.7 min), and 25% (12.2 min) increase in HKDT, respectively, compared to the 10% diet (Figure 4b and Appendix A). Additionally, male flies on the control diet and the 25% diet at age of 12 days had a 35% (14.8 min) and 47% (24.6 min) increase in HKDT, respectively, compared to the 10% diet (Figure 4b and Appendix A).

### 3.5. Composite Fitness Score

To assess whether assembling the data showed the same trends as for the separate analysis of dry weight, lipid content, activity, and HKDT, an ANOVA test was performed. In this analysis, it was assumed that higher values are beneficial for all traits. This test included sex, diet, age, and all the interaction effects. A significant difference between females and males was shown, with females having a 24% (0.15 score) higher fitness score than males (*p* < 0.001, Appendix A). The effect of age and the interaction effect between age and sex were also significant (Appendix A). A significant effect of diet was observed (*p* < 0.001, Appendix A), and the interaction effects, diet and sex (*p* = 0.039, Appendix A), and diet and age (*p* < 0.001, Appendix A) were significant. Additionally, there was also an effect of diet for female and male flies individually (Appendix A). Age and the interaction effect between diet and age affected the fitness score of female flies (Appendix A), while only age, but not the interaction effect between diet and age, affected the fitness score of male flies (Appendix A).

Female flies on the control diet at the age of 12 days had a 20% (0.16 score) and 56% (0.46 score) decrease in fitness score, respectively, compared to the 25% and 10% diets (Figure 5a and Appendix A). Additionally, female flies on the 50% diet at the age of 12 days had a 49% (0.35 score) decrease in fitness score compared to the 10% diet (Figure 5a and Appendix A). Furthermore, female flies on the 25% diet at the age of 4 days had a 20% (0.13 score) decrease in fitness score compared to the 10% diet (Figure 5a and Appendix A). Female flies on the 25% diet at the age of 12 days had a 45% (0.29 score) decrease in fitness score compared to the 10% diet (Figure 5a and Appendix A). Male flies on the control diet, 50% diet, and 25% diet at the age of 12 days had a 44% (0.23 score), 43% (0.23 score), and 39% (0.19 score) higher fitness score, respectively, compared to the 10% diet (Figure 5b and Appendix A).

## 4. Discussion

The natural phenomenon of aging is a focal point in biology and medical science. Aging is characterized by a progressive physiological decline that can be observed in the form of reduced motility, muscle mass, and robustness towards stressors, such as temperature and oxidative stresses, which culminates in increased susceptibility to diseases and eventually death [17,20,21,22,55,56]. Malnutrition is one of several factors that affect the aging process of an individual and typically involves excesses, deficiencies, or an imbalance in the intake of nutrients, consequently leading to either undernutrition or overnutrition. In humans, excess caloric intake is associated with a range of common cardio–metabolic diseases [57], which is a growing concern as a doubling in the number of obese people in the Western World has been observed since 1975 [58]. However, undernutrition and its consequences have been less illuminated even though undernutrition affects the well-being, life- and healthspan of ca. 10% of the human population [59]. The negative impact of malnutrition increases in elderly people and can lead to, e.g., reduced wound healing, performance, and well-being, and cause a weakening of the immune system and premature death [5,9]. Energy expenditures for normal activities are, furthermore, higher in old people (especially men), which means a caloric-deficient diet results in less physical activity, loss of muscle mass, and more illnesses [59]. The incidence of undernutrition in young kids under 5 years of age is also higher for boys than for girls, and this difference between sexes increases with increasing age [60].

In this study, we investigated how malnutrition affected aging in the model organism *D. melanogaster* and to what extent the phenotypic consequences were sex-dependent. Overall, we observed a sex-dependent phenotypic effect on all the investigated traits and when combining data on traits measured on the same individuals (Figure 5). Male flies were generally more sensitive to reduced nutritional availability compared to females. Thus, males suffered from decreased lifespan at lower levels of nutritional stress compared to females. Several other *D. melanogaster* studies have shown decreased lifespan of both female and male flies with decreasing caloric intake [37,42,44], but the strong sex-dependent effect of undernutrition detected in our study is not a general finding. On the contrary, numerous studies using *D. melanogaster* have revealed a positive effect on lifespan with reduced caloric intake [38,39,40,41,42,43]. There can be several reasons for the discrepancy in the observed phenotypic effect of undernutrition observed across *D. melanogaster* studies. First, the medium used in the current study was based on oat, yeast, and sugar, which contrasts with most other studies that only utilize sugar and yeast [38,39,40,41,42]. This leads to different protein-to-carbohydrate ratios and protein quality [61], which ultimately can affect the longevity of *D. melanogaster* [62,63,64]. Second, the above-mentioned studies used water to dilute the diet instead of α-cellulose, as used in the current study. Potentially, this can cause the flies in the present study to be well-hydrated on the control diet and become dehydrated with increasing α-cellulose content (i.e., increasing dilution), as cellulose binds water very effectively [65].

Healthspan provides important additional measures of the aging process as it captures the period with relatively healthy aging and the period with disabilities and age-related diseases [17]. In humans, decreasing locomotor activity is associated with increasing age and decreasing health status, which is also a phenomenon observed in many other species [20,21,22,23,24]. Healthspan also involves robustness towards stressors such as oxidative stress and heat and cold stress [22,25]. The data presented in the current study unambiguously showed that healthspan did not follow the same trajectory as lifespan. Lifespan had already decreased when the flies were exposed to the 50% diet, while it was not until exposure to the 10% diet that locomotor activity and HKDT (both closely associated with healthspan) of the flies decreased. Nutritional conditions, thus, need to be more stressful/extreme to induce a reduction in healthspan compared to the level of nutritional stress needed to reduce lifespan. This suggests that flies utilize all available nutrients to maintain processes that allow for physiological functions and stress resistance at the expense of lifespan.

Female flies were able to maintain better lifespan and healthspan at a lower nutritious diet than male flies, suggesting that female flies were better able to cope with low nutritional environments (Figure 5). Interestingly, females displayed a decrease in dry weight and lipid content with increased undernutrition, whereas males only displayed reduced dry weight and not reduced lipid content when nutrients were scarce. Such disparity suggests that female flies, to a larger extent than males, utilize lipid content to ensure physiological functions. Lipids are important for reproduction, which suggests that female flies prioritize these traits at extremely low-calorie diets, while the opposite is true for males [66,67].

Our data suggest that male flies were more sensitive to undernutrition than female flies and that the consequences on health- and lifespan were more severely impacted in males compared to females. In an evolutionary context, the female sex typically invests more in reproductive traits and, thus, may have evolved to better cope with nutritional stress. Further, female insects are often better at coping with diverse environmental stressors [27,68], which can contribute to skewed sex ratios in natural populations with excess female individuals. Relatively more females in a population might be beneficial as it increases the potential number of offspring produced. However, the skewed sex ratios can also cause a reduction in effective population sizes, which can lead to increased rates of inbreeding and loss of genetic variation [69]. Thus, in this context, our results are also an important contribution.

On most of the diets tested in our study, female flies showed an increase in locomotor activity, heat stress tolerance, body weight, and lipid content from 4 days of age to 12 days of age. An increase in dry weight and lipid content early in life in female flies was also found by Djawdan et al. [70] in female flies with a high lifespan and low fecundity and not in females with a short lifespan and high fecundity. It can, therefore, be speculated that the population used in our study allocated resources into lifespan extension that increases survival at the cost of reproduction. The increase in locomotor activity in young flies was followed by a decrease and this has also been observed by others [71,72,73]. However, Fernández et al. [74] found that locomotor activity differed across populations, suggesting that absolute healthspan and lifespan, can be difficult to compare directly across studies, unless the exact same populations and environments are used.

In conclusion, we found that undernutrition had severe negative impacts on lifespan, lipid content, dry weight, locomotor activity, and heat stress tolerance in *D. melanogaster* and that these negative impacts of undernutrition were more severe in males than in females. Thus, the effects of undernutrition on fitness components are highly sex-dependent. Further, we found that healthspan seems to be less negatively impacted by nutritional stress compared to lifespan, which might be explained by healthspan being more closely linked to fitness, as increased healthspan may lead to an expansion of the period where reproduction occurs.

## Figures and Tables

**Figure 1 insects-15-00009-f001:**
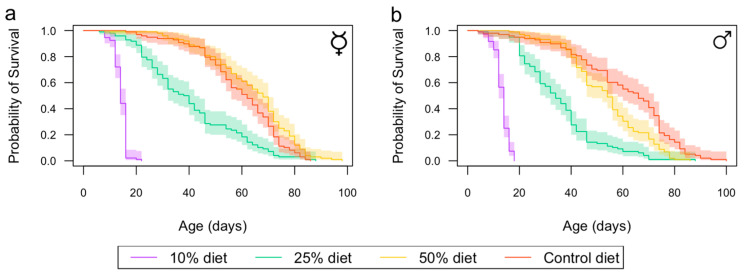
The effects of control (red) and 50% (yellow), 25% (green), and 10% (purple) diets on female and male lifespan. Each Kaplan–Meier survival curve contains *n* = 100 individuals. Kaplan–Meier survival curves are shown for females (**a**) and males (**b**) (shaded areas are 95% confidence intervals). The ☿ sign indicates that females were virgins.

**Figure 2 insects-15-00009-f002:**
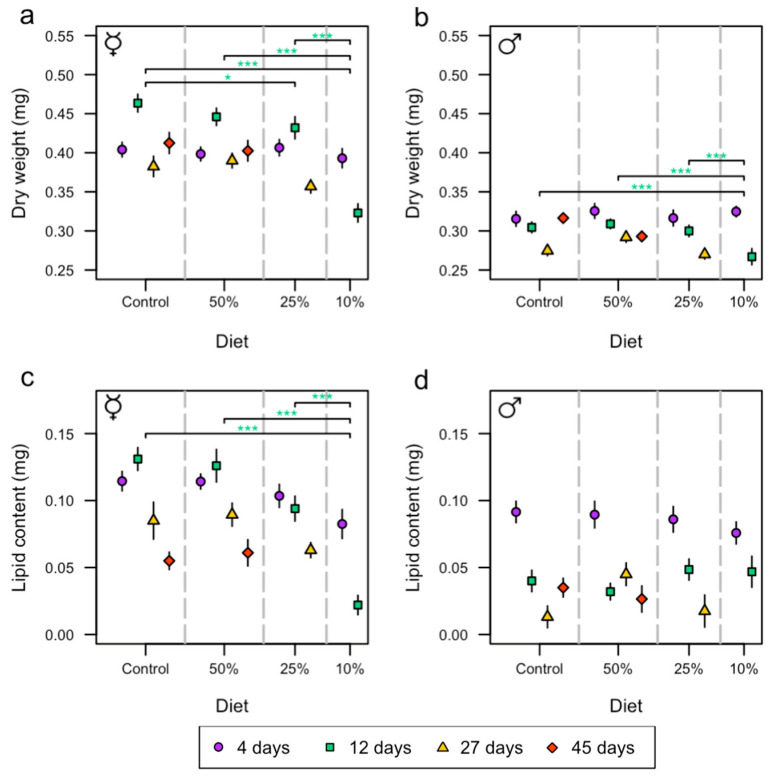
The effects of diet (control, and 50%, 25%, and 10%) on female and male dry weight and lipid content at the age of 4 (purple circle), 12 (green square), 27 (yellow triangle), and 45 days (red diamond). Each point contains *n* = 20 individuals. Asterisks (* *p* < 0.05; *** *p* < 0.001) indicate statistically significant differences in pairwise comparisons between diets where the colors represent the test day. Dry weight of female (**a**,**b**) male flies (mean ± SE). Lipid content of females (**c**,**d**) males of different ages (mean ± SE). The ☿ sign indicates that females were virgins.

**Figure 3 insects-15-00009-f003:**
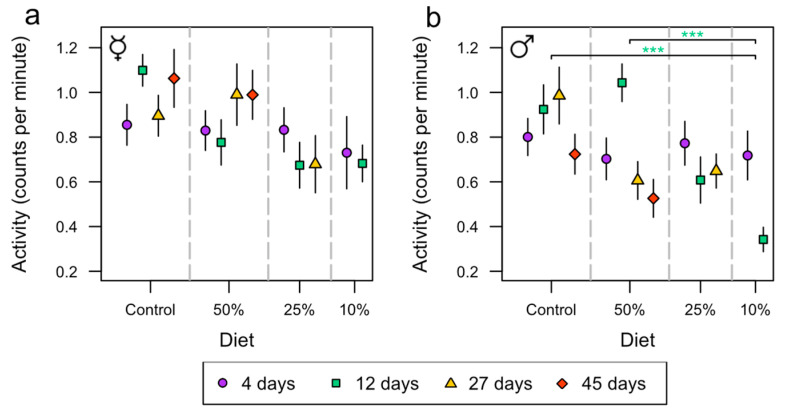
The effects of diet (control, 50%, 25%, and 10%) on male and female activity at the age of 4 (purple circle), 12 (green square), 27 (yellow triangle), and 45 days (red diamond). Each point contains *n* = 20 individuals. Asterisks (*** *p* < 0.001) indicate statistically significant pairwise differences between diets where the colors represent the test day. Activity, measured in counts per minute, of female (**a**,**b**) male flies (mean ± SE). The ☿ sign indicates that females were virgins.

**Figure 4 insects-15-00009-f004:**
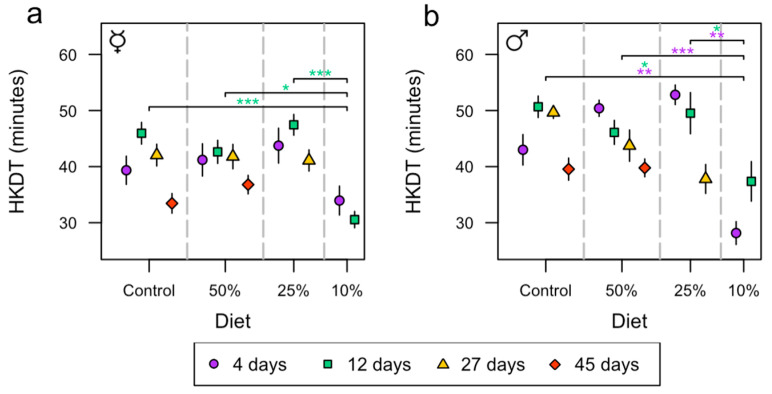
The effects of diet (control, and 50%, 25%, and 10%) on male and female flies’ heat knockdown time (HKDT) at the age of 4 (purple circle), 12 (green square), 27 (yellow triangle), and 45 days (red diamond). Each point contains *n* = 20 individuals. Asterisks (* *p* < 0.05; ** *p* < 0.01; *** *p* < 0.001) indicate statistically significant differences among diets where the colors represent the test day. HKDT of female (**a**,**b**) male flies (mean ± SE). The ☿ sign indicates that females were virgins.

**Figure 5 insects-15-00009-f005:**
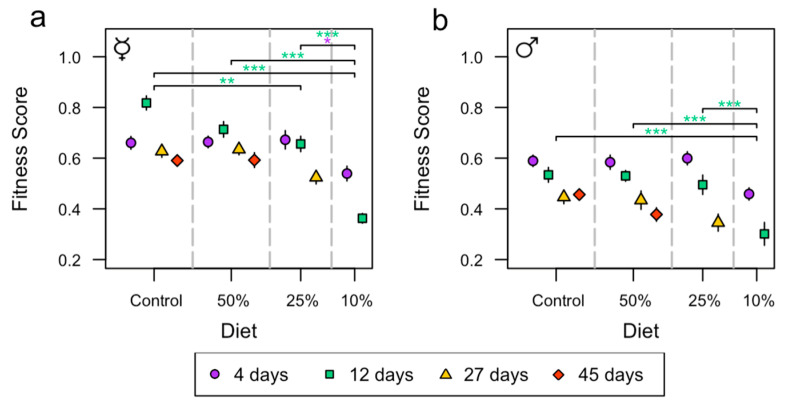
The effects of diet (control, and 50%, 25%, and 10%) on assembled data of dry weight, lipid content, activity, and HKDT of male and female flies at the age of 4 (purple circle), 12 (green square), 27 (yellow triangle), and 45 days (red diamond). For all traits, high values are assumed to be beneficial. Each point contains *n* = 20 individuals. Asterisks (* *p* < 0.05, ** *p* < 0.01; *** *p* < 0.001) indicate statistically significant differences between pairwise comparisons among diets where the colors represent the test day. Assembled data of dry weight, lipid content, activity, and HKDT of female (**a**,**b**) male flies (mean ± SE). The ☿ sign indicates that females were virgins.

**Table 1 insects-15-00009-t001:** Nutritional components of the four different diet types. The control diet (standard Leeds) was set to have 100% nutritional value, and nutritional values in the remaining three diet types were set relative to that.

Diet Types	Yeast	Sucrose	Oat	Agar	Nipagin	Acetic Acid	Cellulose
Control diet	60 g	40 g	30 g	16 g	12 mL	1.2 mL	0 g
50%	30 g	20 g	15 g	16 g	12 mL	1.2 mL	65 g
25%	15 g	10 g	7.5 g	16 g	12 mL	1.2 mL	97.5 g
10%	6 g	4 g	3 g	16 g	12 mL	1.2 mL	117 g

**Table 2 insects-15-00009-t002:** Chronological age of the flies when tested in the locomotor activity assay and heat stress tolerance assay. Green indicates flies that were used at a specific chronological age, and red shows when no flies were available for testing. The control diet (standard Leeds) is the optimal diet, and 50%, 25%, and 10% denote the diluted diets.

		Diet Types			
		Control	50%	25%	10%
**Age**	4 days				
	12 days				
	27 days				
	45 days				

## Data Availability

All data generated and/or analyzed during the current study are available from the corresponding author upon request.

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
