# Peer review of "Strong Sex-Dependent Effects of Malnutrition on Life- and Healthspan in Drosophila melanogaster"

_insects, 2023, doi:10.3390/insects15010009_

Round 1

Reviewer 1 Report

Comments and Suggestions for Authors

Abstract

Informative, well prepared

Introduction

The work is well presented and justified.

Material and methods

line 118: Nipagen - Nipagin

line 168: passed the laser - crossed the infrared beam

line 190: Multivariate Cox regression was used to determine the difference in lifespan with sex and diet as independent variables - This data (hazard rates) should be presented in the Results section.

lines 203-205: Please provide more detail information about calculation of Composite fitness score.

2.7. Statistical analysis: What software was used to create the illustrations and statistical processing?

Results

Results from Cox regression analysis should be reported.

Discussion

Well prepared, informative and based on adequate references.

References

Adequate and up-to-date.

Author Response

Dear Reviewer,

We thank you for constructive and positive evaluations of our work. In our re-submitted manuscript we have addressed all the comments raised. All modifications in the manuscript have been performed with track change. We hope that our detailed responses below, and the changes made with the manuscript meets the standards of the journal allowing us to publish the work in Insects.

Nikolaj Klausholt Bak*, Torsten Nygaard Kristensen* and Palle Duun Rohde**.

Department of Chemistry and Bioscience* and Department of Health Science and Technology**.

Aalborg University

Abstract
Informative, well prepared
Response: Thank you.

 Introduction
The work is well presented and justified. 
Response: Thank you.

Material and methods
line 118: Nipagen – Nipagin
Response: This has been corrected.

line 168: passed the laser - crossed the infrared beam
Response: This has been corrected.

line 190: Multivariate Cox regression was used to determine the difference in lifespan with sex and diet as independent variables - This data (hazard rates) should be presented in the Results section.
Response: A Supplementary Table (Table S1) containing hazard rates have now been included in the Supplementary Material, and a reference to the table has been included in the manuscript (see line 279).

lines 203-205: Please provide more detail information about calculation of Composite fitness score.
Response: More detailed information about calculation of composite fitness score has been added to lines 237-243 and 263-272 in the revised manuscript.

2.7. Statistical analysis: What software was used to create the illustrations and statistical processing?
Response: We used R version 4.3.1 for figures and statistical analysis, as well as the ‘Survival’ package in R for Multivariate Cox regression. This has been added to the manuscript (lines 234-236).

Results
Results from Cox regression analysis should be reported.
Response: As mentioned above, a table (Table S1) containing hazard rates has now been included in the Supplementary Material 1, and a reference to the table in the manuscript line 279.

 Discussion
Well prepared, informative and based on adequate references.
Response: Thank you.

References
Adequate and up-to-date.
Response: Thank you.

Reviewer 2 Report

Comments and Suggestions for Authors

This manuscript investigates the effects of malnutrition on various fitness components in Drosophila melanogaster, using four diets with differing nutritional values. The manuscript is well written, the methodology of this study is well-conceived, and the findings are valid and insightful. I have only a few comments and suggestions to further enhance the clarity of the study.

In the introduction, while the hypotheses regarding the effects of undernutrition on Drosophila melanogaster are clearly stated, they could benefit from a more explicit linkage to existing literature. Elaborating on the scientific basis for expecting a lifespan increase with moderate nutrient reduction, and the rationale behind males being more negatively impacted by malnutrition, would make the hypotheses more informative.

In addition, it would be beneficial to explicitly state what novel contributions this study makes to the field and what aspects of your research are new or different from previous studies.

The manuscript describes the experimental diets as being diluted with α-cellulose to create different nutritional environments (50%, 75%, 90%). The current naming could lead to confusion regarding the actual nutritional content of each diet. In my opinion, it would be more informative to rename the diets in a way that reflects their nutritional content relative to the control diet (standard Leeds, which is set at 100%). This approach will provide clearer information about the relative nutritional value of each diet.

This is just a remark. In this study, activity was monitored for a 6-hour period, which offers a useful snapshot but a longer observation window, spanning various times of the day, could provide a more comprehensive understanding of how different nutritional conditions impact locomotor activity.

Although the statistical methods for analyzing healthspan are clearly outlined, the manuscript would benefit from a more explicit definition of how 'healthspan' is defined and measured in this study. Specifically, it would be helpful to understand how locomotor activity and heat knock-down time are used as indicators of healthspan in Drosophila. For example, are there specific thresholds or patterns in these measures that the authors consider indicative of good or declining health?

Author Response

Dear Reviewer,

We thank you for constructive and positive evaluations of our work. In our re-submitted manuscript we have addressed all the comments raised. All modifications in the manuscript have been performed with track change. We hope that our detailed responses below, and the changes made with the manuscript meets the standards of the journal allowing us to publish the work in Insects.

Nikolaj Klausholt Bak*, Torsten Nygaard Kristensen* and Palle Duun Rohde**.

Department of Chemistry and Bioscience* and Department of Health Science and Technology**.

Aalborg University

This manuscript investigates the effects of malnutrition on various fitness components in Drosophila melanogaster, using four diets with differing nutritional values. The manuscript is well written, the methodology of this study is well-conceived, and the findings are valid and insightful. I have only a few comments and suggestions to further enhance the clarity of the study.

In the introduction, while the hypotheses regarding the effects of undernutrition on Drosophila melanogaster are clearly stated, they could benefit from a more explicit linkage to existing literature. Elaborating on the scientific basis for expecting a lifespan increase with moderate nutrient reduction, and the rationale behind males being more negatively impacted by malnutrition, would make the hypotheses more informative.

Response: This has been added to the manuscript (lines 101-121).

In addition, it would be beneficial to explicitly state what novel contributions this study makes to the field and what aspects of your research are new or different from previous studies.

Response: Novel contributions of this study has been added to the manuscript (please see lines 121-126 and 130-136).

The manuscript describes the experimental diets as being diluted with α-cellulose to create different nutritional environments (50%, 75%, 90%). The current naming could lead to confusion regarding the actual nutritional content of each diet. In my opinion, it would be more informative to rename the diets in a way that reflects their nutritional content relative to the control diet (standard Leeds, which is set at 100%). This approach will provide clearer information about the relative nutritional value of each diet.

Response: This has been changed throughout the revised manuscript.

This is just a remark. In this study, activity was monitored for a 6-hour period, which offers a useful snapshot but a longer observation window, spanning various times of the day, could provide a more comprehensive understanding of how different nutritional conditions impact locomotor activity.
Response: Thank you for the remark.

Although the statistical methods for analyzing healthspan are clearly outlined, the manuscript would benefit from a more explicit definition of how 'healthspan' is defined and measured in this study. Specifically, it would be helpful to understand how locomotor activity and heat knock-down time are used as indicators of healthspan in Drosophila. For example, are there specific thresholds or patterns in these measures that the authors consider indicative of good or declining health?

Response: This is a very good question, unfortunately we do not operate with specific thresholds of healthspan in gerontology, we can only talk about relative sizes of healthspan held up against each other. This is because the traits we use to measure healthspan depend on the genetics of the population investigated, environments and possible interaction between genes and environments. When we measure a fly in our laboratory, its healthspan relative to its environment will be average let’s say when it has an activity of 0.9 counts per minute. But this can both be an extremely low or high value compared to other fly populations. The same phenomenon applies when we measure lifespan, here there is also a significant difference between populations, even when exposed to the same environment. Some live 40 days, others 80 days. We have now addressed this in the manuscript lines 519-523.

Reviewer 3 Report

Comments and Suggestions for Authors

Animals can develop many related diseases under malnutritional conditions. Aging is also able be affected by malnutritional conditions. This manuscript investigated the effects of malnutrition on both aging and nutrition-related phenotypes including body weight, locomotor activity, heat stress tolerance, and fat content with sex specificity to some extent. This research can help advance our understanding of how undernutrition plays a role in aging and aging-related life processes about healthspan. Overall, this investigation is of interest to readership. The manuscript was written well. However, some issues should be addressed before it can be accepted for publication. I have some comments below, expecting help improve the manuscript. 

Major comments:

1. For all female flies, according to the methods part, they are denoting virgin females. But in all Figures, female is indicated with â™€ï¸Žthat is for mated femaleswhich is confusing.

2. Authors mainly showed the data at the days of 12 Figures 2, 3,4 and 5. It is obvious that nearly all females with the control food (sometime for other food like 50%, 75%) showed the highest value for these figures at the days of 12 compared to other days like 4, 27, and 45 days but males did not. It seems that the age at 12 days is a special time point. Do authors have any hypothesis about this phenomenon?

3. For the very old flies at the day of 45 days, they were only tested for control and 50% diluted food. Results seem suggesting that there is no difference between two group tests for all figures except Figure 3b. If this is the case, this part of results seem unnecessary.

4. In Figure 1, what is the unit of the horizontal axis? Authors should make it clear to make it more easily understandable. The boxplot at the bottom is also confused, what is the purpose? I did not understand what this boxplot wants to indicate. Additionally, in the figure legend, authors indicated “Boxplot outliers is indicated by circles and is outside of the range from (Q1-1.5×IQR) to (Q3+1.5×IQR)”. But what is (Q1-1.5×IQR) to 221 (Q3+1.5×IQR)? Authors should explain them somewhere, for example in the methods part. 

5. Through the text, many data about the “interaction effects” were showed, although I can understand it, it will be helpful if authors can give some explanation because there is no such explanation in the manuscript. 

6. For Figure 2d, although the significance is not so obvious as female, authors should also indicate the figure by showing the word “Figure 2d” in the text because it is also the main data. I did not find any information about “Figure 2d” in the text. It will be confused if the Figure 2d is missing when reading the manuscript. Similar case for Figure 3a.

Minor comments:

1. lines 216-217, Figure should be Figures.

2.  the P should be italic all over the manuscript. 

3. For example Lines 239,24,  the format about indication of Figure in the bracket should be keeping same as lines 216-217. They should be same though the manuscript. 

Author Response

Dear Reviewer,

We thank you for constructive and positive evaluations of our work. In our re-submitted manuscript we have addressed all the comments raised. All modifications in the manuscript have been performed with track change. We hope that our detailed responses below, and the changes made with the manuscript meets the standards of the journal allowing us to publish the work in Insects.

Nikolaj Klausholt Bak*, Torsten Nygaard Kristensen* and Palle Duun Rohde**.

Department of Chemistry and Bioscience* and Department of Health Science and Technology**.

Aalborg University

Animals can develop many related diseases under malnutritional conditions. Aging is also able be affected by malnutritional conditions. This manuscript investigated the effects of malnutrition on both aging and nutrition-related phenotypes including body weight, locomotor activity, heat stress tolerance, and fat content with sex specificity to some extent. This research can help advance our understanding of how undernutrition plays a role in aging and aging-related life processes about healthspan. Overall, this investigation is of interest to readership. The manuscript was written well. However, some issues should be addressed before it can be accepted for publication. I have some comments below, expecting help improve the manuscript. 

Major comments:

  1. For all female flies, according to the methods part, they are denoting virgin females. But in all Figures, female is indicated with ♀︎that is for mated females, which is confusing.
    Response: We are using virgin females and we have therefore corrected the female symbol in figures 1-5.

  1. Authors mainly showed the data at the days of 12 Figures 2, 3,4 and 5. It is obvious that nearly all females with the control food (sometime for other food like 50%, 75%) showed the highest value for these figures at the days of 12 compared to other days like 4, 27, and 45 days but males did not. It seems that the age at 12 days is a special time point. Do authors have any hypothesis about this phenomenon?
    Response: This has been addressed in lines 513-523. It was hypothesised that the early increase in fat content and body weight might be due to allocation of resources to lifespan extension and other physiological components that increase survival at the expense of reproduction.

  1. For the very old flies at the day of 45 days, they were only tested for control and 50% diluted food. Results seem suggesting that there is no difference between two group tests for all figures except Figure 3b. If this is the case, this part of results seem unnecessary.
    Response: This is a good point, and this would definitely highlight the main results, however, we argue that this visually illustrates that even though the flies on the control, 50% and 25% diets have lived longer than the flies on the 10% diet, they either still have better fitness or at least the same fitness as flies on the 10% diet.

  1. In Figure 1, what is the unit of the horizontal axis? Authors should make it clear to make it more easily understandable. The boxplot at the bottom is also confused, what is the purpose? I did not understand what this boxplot wants to indicate. Additionally, in the figure legend, authors indicated “Boxplot outliers is indicated by circles and is outside of the range from (Q1-1.5×IQR) to (Q3+1.5×IQR)”. But what is (Q1-1.5×IQR) to 221 (Q3+1.5×IQR)? Authorsshould explain them somewhere, for example in the methods part.
    Response: We see that the boxplot does not contribute with new information that the Kaplan Meier curves does not already show, and therefore the boxplots have been removed.

  1. Through the text, many data about the “interaction effects” were showed, although I can understand it, it will be helpful if authors can give some explanation because there is no such explanation in the manuscript.
    Response: Explanation of “interaction effects” have now been included in the methods part of the manuscript – please see lines 244-253.

  1. For Figure 2d, although the significance is not so obvious as female, authors should also indicate the figure by showing the word “Figure 2d” in the text because it is also the main data. I did not find any information about “Figure 2d” in the text. It will be confused if the Figure 2d is missing when reading the manuscript. Similar case for Figure 3a.
    Response: Suggestion has been followed – please see figure 2d and 3a and lines 310 and 336.

Minor comments:

  1. lines 216-217, Figure should be Figures.
    Response: This has been corrected.

  1. the P should be italic all over the manuscript.
    Response: This has been corrected throughout the manuscript.

  1. For example Lines 239,24,  the format about indication of Figure in the bracket should be keeping same as lines 216-217. They should be same though the manuscript.
    Response: This has been corrected throughout the manuscript.

Round 2

Reviewer 3 Report

Comments and Suggestions for Authors

Authors addressed my concerns well. I have no further comments. I advise that this manuscript can be accepted for publication.